

**Permafrost thawing exhibits a greater influence on bacterial richness**
**and community structure than permafrost age in Arctic permafrost**
**soils**
Mukan Ji[1, 2], Weidong Kong[1,2,3]*, Chao Liang[4], Tianqi Zhou[1,2], Hongzeng Jia[1,2], Xiaobin Dong[5]
[1]Key Laboratory of Alpine Ecology, Institute of Tibetan Plateau Research, Chinese Academy of
Sciences (CAS), Beijing 100101, China
[2]College of Resources and Environment, University of Chinese Academy of Sciences, Beijing 100039,
China
[3]CAS Center for Excellence in Tibetan Plateau Earth Sciences, Chinese Academy of Sciences, Beijing
100101, China.
[4]Institute of Applied Ecology, Chinese Academy of Sciences, Shenyang, 110016, China
[5]State Key Laboratory of Earth Surface Processes and Resource Ecology, College of Resources Science
and Technology, Beijing Normal University, Beijing 100875, China
Corresponding author *:
Institute of Tibetan Plateau Research, Chinese Academy of Sciences, Building 3, Courtyard 16, Lincui
Road, Chaoyang District, Beijing 100101, China.
Phone: 8610–84097039; Fax: 8610–84097079; E–mail: wdkong@itpcas.ac.cn
Running title: Relative influences of permafrost thawing and age on soil bacteria



**Abstract**
Global warming accelerates permafrost thawing and changes permafrost microbial community
structure, but little is known about how microorganisms in permafrost with different ages respond to
thawing. Herein, we disentangled the relative importance of permafrost age (young, medium, old, and
ancient) spanning from 50 to 5,000 yr and thawing status (active, transitional, and permanently frozen)
in shaping bacterial community structure using Hiseq sequencing of the 16S rRNA gene. Our results
revealed significant influences of both permafrost thawing and age on bacterial richness. The bacterial
richness was significantly higher in the young and thawed permafrost, and the richness increase was
mainly observed in *Firmicutes*, *Actinobacteria*, *Chloroflexi*, *Deltaproteobacterai*, and
*Alphaproteobacteria*. Permafrost thawing led to a gradual change in bacterial community structure and
increased the contribution of determinism to shape the bacterial community assembly. Permutational
analysis of variance demonstrated that thawing significantly changed bacterial community structure at
all soil ages, but the community convergence due to permafrost thawing was not observed. Structural
equation modeling revealed that permafrost thawing exhibited a greater influence on both bacterial
richness and community structure than permafrost age. Our results indicate that microorganisms in
permafrost with different ages respond differently to thawing, which eventually leads to distinct
bacterial community compositions and different soil organic carbon degradation processes during
permafrost thawing.
**Keywords:** Permafrost thawing; permafrost age; bacterial community; richness; Arctic



## 1 Introduction

Global warming accelerates permafrost thawing, and 200 billion tons of carbon is estimated to be released into the atmosphere from global permafrost over the next 300 years (Turetsky et al., 2019). The degradation of soil organic carbon (SOC) is predominately driven by microorganisms (Frank–Fahle et al., 2014), and the quality and quantity of SOC also control the abundance and community structure of microbial community (Chen et al., 2016). It has been reported that permafrost of different thawing status and ages exhibits distinct labile and recalcitrant carbon quantities, with a higher carbohydrates in relation to aliphatic carbon in older than in younger permafrost (Chen et al., 2016; Mueller et al., 2015; Yang et al., 2009). Thus, the distinct SOC composition may subsequently impact the microbial community structure in permafrost soil, and the distinct microbial community structure may respond differently to permafrost thawing. However, the impacts of permafrost age and its interaction with thawing on microbial community remain largely elusive.

In addition to permafrost age, contrasting structure of labile and recalcitrant carbons was also reported in the thawed and frozen permafrost. This was proposed to be due to the distinct microbial transformation process in the different permafrost thawing status (Mueller et al., 2015). The microbes in the frozen permafrost are predominately in a state with reduced metabolism rate (Gilichinskii, 1995), thus labile carbon is protected from microbial degradation (Hobbie et al., 2000). In contrast, permafrost thawing substantially activates a diverse range of oligotrophic and copiotrophic bacteria, and enriches carbohydrate transporter and metabolism–related genes (Schostag et al., 2019). This leads to an increased bacterial richness and converged community metabolic functions, and the soil carbon being dominated by aliphatic carbon resulted from microbial transformation (Deng et al., 2015; Mackelprang et al., 2011; Monteux et al., 2018; Schostag et al., 2019).

Soil development leads to changes in bacterial community structure, predominately due to nutrient accumulation and vegetation colonization (Bardgett and Walker, 2004; Park et al., 2011). Distinct bacterial community structure has been reported in soils of different ages. For example, young soils in the deglaciation chronosequence exhibit significantly lower bacterial richness than aged soils, and autotrophs play a major role in the accumulation of nutrients (Kazemi et al., 2016; Kim et al., 2017; Liu et al., 2016). In contrast, aged soils with vegetations are dominated by heterotrophs, such as *Acidobacteria* and *Actinobacteria* (Kwon et al., 2015). However, little is known about the influence of





permafrost age on soil microbial community.
To explore the effects of permafrost age on the response of bacteria to permafrost thawing, soil
bacterial community in Arctic permafrost was characterized using the Illumina sequencing targeting the
16S rRNA gene. Given the continuously changed bacterial community with increasing soil age
(Kazemi et al., 2016; Kim et al., 2017), we hypothesized that bacterial richness and community
structure would also significantly differ in the permafrost of various ages and response differently to
permafrost thawing. Permafrost in northern Alaska varies in age (Hinkel et al., 2003), and thus provides
a perfect opportunity to investigate the influence of permafrost thawing status and age on the
permafrost soil bacterial community.
**2 Materials and methods**
*2.1 Site description*
The permafrost was sampled in the Barrow Peninsula between 71°20′to 71°27′N latitude and between
156°4′and 156°7′W longitude (Kao–Kniffin et al., 2015). Barrow Peninsula is located at the
northernmost coast of Alaska, and is part of the Arctic Coastal Plain with continuous permafrost. The
mean annual temperature is -12°C, and the mean annual precipitation is 104 mm (Mueller et al., 2015).
In brief, soil cores were collected along a chronosequence of drained lake basin, spanning in age from
young (< 50 yr old), medium (< 300 yr old), old (< 3,000 yr old), to ancient (3,000–5,000 yr old) in
April 2010. The chronosequence was determined by the degree of plant community succession and $^{14}$C
carbon dating (Hinkel et al., 2003). At each lake basin, a soil core was collected using a SIPRE corer
measuring 80 to 150 cm long and 7.5 cm diameter attached to a Big Beaver earth drill apparatus (Litter
Beaver, Inc., Livingstone, TX, USA) mounted on a sledge. Each soil core contained three layers: active,
transition, and permanently frozen. The active layer represents the surface soil layer that thaws and
refreezes on an annual basis; the transition layer remains frozen, but occasionally thaws during warmer
summers; the permanently frozen layer remains annually frozen (Kao–Kniffin et al., 2015). The surface
organic layer thickness vary with permafrost age, which was < 5, 10–15, 15–30, and 40–50 cm for the
young, medium, old, and ancient–aged permafrost soils (Kao–Kniffin et al., 2015). The frozen soil
cores were cut with a chop–saw into sections of corresponding soils horizons in a cold room in Barrow,
and soils were homogenized, stored, and transported at -20 °C until processed (Mueller et al., 2015).



Soil total organic carbon (TOC) and total nitrogen (TN) were measured using dry combustion (Vario
MAX CNS Analyzer, Elementar, Hanau, Germany) (Mueller et al., 2015).
*2.2 DNA extraction and sequencing*
Total DNA was extracted using the MO BIO Power Soil DNA extraction kit (Mo Bio Laboratories,
Carlsbad, CA, USA) according to the manufacturer's instructions. Universal primer set 515F
(5'–GTGCCAGCMGCCGCGGTAA–3') and 806r (5'–GGACTACHVGGGTWTCTAAT–3') with
12–nt unique barcodes was used to amplify the V4 hyper–variable region of the 16S rRNA gene
(Caporaso et al., 2012). The PCR mixture (25 µl) contained 1x PCR buffer, 1.5 mM of $MgCl_2$, 0.4 µM
each of deoxynucleoside triphosphate bases, 1.0 µM of each primer, 0.5 U of Ex Taq (TaKaRa, Dalian,
China) and 20 ng of DNA template. The PCR amplification program included an initial denaturation at
94 °C for 3 min, followed by 30 cycles of 94 °C for 20 s, 56 °C for 30 s, and 72 °C for 45 s, and a final
extension at 72 °C for 10 min. PCR products were pooled in equal molar amounts, and then used for
pair–end sequencing (2 x 250 bp) on the Illumina HiSeq 2500 sequencer at the Magigene (Guangzhou,
China).
*2.3 Data processing*
Three samples generated very low reads, to avoid artefact from different sequencing batches, these
three samples were removed from down steam analysis. Raw sequence data were processed using the
MOTHUR v. 1.34.3 (Schloss et al., 2009). Paired–end reads were merged and quality screened with the
following settings: as the amplicon size was approximately 300 bp, sequences with length < 250 or >
350, more than 1 mismatch in the primer region, average quality < 30, ambiguous bases > 0 and
homopolymer length > 9 were removed from the subsequent downstream analysis. The sequences were
then aligned against the Silva reference alignment (release 128), which was trimmed to include only
the same region amplified, and those sequences that did not align were removed. Chimeric sequences
were identified using the UCHIME (Edgar et al., 2011) and removed. The remaining sequences were
classified using the Bayesian classifier against the Silva database (release 128), with a minimum
confidence score of 80 % (Wang et al., 2007), and all Eukaryota, chloroplasts, mitochondria and
unknown sequences were removed. Archaeal sequences were also removed to concentrate the study on
the bacterial community. Finally, sequences were classified into operational taxonomic units (OTUs) at





the 97 % identity, and singletons were then removed. The dataset was sub–sampled to an equal depth of
16,144, which was the smallest sample size across the entire dataset. Bacterial richness (OTU
recovered) was calculated using the summary.single command in the Mothur program (Schloss et al.,

130 2009).

*2.4 Statistical analysis*
Significant differences in bacterial richness, total organic carbon and total nitrogen across permafrost
age and thawing status were tested using the two–way ANOVA, and the pairwise differences were
assessed by the Tukey's HSD test using SPSS 23 (SPSS Inc., Armonk, NY, USA). The Levene's test
was used to ensure the homogeneity of variances for the dependent variables (bacterial richness, TOC,
and TN) for each combination of the independent variables (Brown and Forsythe, 1974). One–way
ANOVA was used to examine the significance of the differences among the permafrost soils of
different thawing status with the same permafrost age.
Non–metric Multidimensional Scaling (NMDS) was generated from the Hellinger–transformed
bacterial community dataset based on the Bray–Curtis dissimilarity matrix using Primer 6 (Clarke and
Warwick, 2006). The contributions of carbon, nitrogen, C:N ratio, permafrost age, and thawing status
to the community structure were quantified using the distance–based linear model (DistLM) after
normalisation. Permutational analysis of variance (PERMANOVA) was used to examine the influence
of permafrost thawing and age on bacterial community structure (Anderson, 2001) using Primer 6. We
compared the multivariate dispersion homogeneity to assess the bacterial community convergence by
permafrost thawing status, using permutational analysis of multivariate dispersions (PERMDISP)
(Anderson, 2006).
*2.5 Quantifying the contribution of stochasticity*
Bray–Curtis dissimilarity based normalised stochasticity ratio (NST) index was calculated using the
'NST' package in R (http//www.r-project.org) to represent the contribution of stochasticity to
community assembly (Ning et al., 2019). The NST index values range from 0 % to 100 %, a 0 %
indicates zero contribution of stochasticity, whereas 100 % indicates the community assembly being
completely stochasticity–driven.
*2.6 Structural equation modelling (SEM) analysis*



We conducted SEM using AMOS 21 software (IBM SPSS Inc., Chicago, IL, USA) to assess the
relative importance of permafrost thawing status and age in shaping bacterial richness and community
structure. The permafrost age was ranked from 1 to 4 for the youngest to the oldest permafrost soils,
whereas the thawing status was ranked from 1 to 3 for the active to frozen permafrost. The standardised
regression weights were calculated for the bacterial richness and the $1^{st}$ and $2^{nd}$ axis coordinates of the
NMDS ordination plot. The goodness of fit for the model was judged by the following measures (Guo
et al., 2015): (1) comparative fit index (CFI, the model has a good fit when $0.97 \leq CFI \leq 1.00$, and an
acceptable fit when $0.95 \leq CFI < 0.97$); (2) goodness–of–fit index (GFI, the model has a good fit when
$0.95 \leq GFI \leq 1.00$, and acceptable fit when $0.90 \leq GFI < 0.95$); (3) normed fit index (NFI, the model
has a good fit when $0.95 \leq NFI \leq 1.00$ and an acceptable fit when $0.90 \leq NFI < 0.95$); (4) $\chi^2$ test; the
model has a good fit when $0 \leq \chi^2 / d.f. \leq 2$ and $0.05 < P \leq 1.00$, and an acceptable fit when $2 < \chi^2 / d.f. \leq$
3 and $0.01 \leq P \leq 0.05$); and (5) the root mean square error of approximation (RMSEA, the model has a
good fit when $0 \leq RMSEA \leq 0.05$ and $0.10 < P \leq 1.00$, and an acceptable fit when $0.05 < RMSEA \leq$
0.08 and $0.05 \leq P \leq 0.10$).
**3 Results**
*3.1 The influence of permafrost age and thawing status on soil organic carbon and nitrogen*
Across all samples, soil total organic carbon (TOC) ranged from 0.5 % to 35.6 %, and exhibited
significant differences by permafrost thawing status (Two–way ANOVA, $P < 0.01$, Fig. 1a), but not by
permafrost age ($P = 0.343$, Fig. 1b). The active layer soil exhibited the highest TOC (16.7 %), and was
significantly higher than the permanently frozen layer soil (5.6 %, Tukey's HSD $P < 0.001$). Soil total
nitrogen (TN) ranged from 0.1 % to 1.5 %, and significant differences were only detected by
permafrost thawing status ($P = 0.007$, Fig. 1c), but by permafrost age ($P = 0.446$, Fig. 1d). The active
layer soil exhibited the highest TN (0.73 %), and was significantly higher than the permanently frozen
layer soil (0.29 %, Tukey's HSD, $P = 0.004$).
*3.2 The influence of permafrost age and thawing status on bacteria richness*
A total of 1,679,607 bacterial sequences were retained, with an average sequence length of 292 bp.
There were 2,659 OTUs identified at the 97 % nucleic acid sequence identity. After rarefying to an



equal depth, 2,415 bacterial OTUs were retained, and the community was dominated by *Firmicutes*
(42 %), *Actinobacteria* (28.9 %), and *Proteobacteria* (10.6 %, Supplementary Fig. 1).
Our results exhibited substantial differences in the bacterial richness among the permafrost soils of
different thawing status (Two–way ANOVA, $P < 0.001$; Fig. 2a, Supplementary Table 1) and ages ($P =$
0.013; Fig. 2b). A significantly higher bacterial richness was observed in the active layer soil (358
OTUs) than the transition (287 OTUs; Pairwise Tukey's HSD tests, $P = 0.011$) and the frozen layer
soils (248 OTUs, $P < 0.001$, Supplementary Table 2). Young permafrost soil (380 OTUs) exhibited a
significantly higher bacterial richness than the medium (265 OTUs, $P = 0.001$), old (287 287, $P =$
0.002), and ancient soils (271 OTUs, $P = 0.009$, Supplementary Table 3).
Within each age category, the significant influence of permafrost thawing was only observed in the
young permafrost soil (one–way ANOVA, $P < 0.001$, Fig. 2b, Supplementary Table 4), whereas those
in the medium, old, and ancient soils were non–significant ($P = 0.445$, 0.48, and 0.35, respectively). In
the young permafrost soil, permafrost thawing significantly increased OTU number from 248 in the
frozen layer soil to 471 in the active layer soil (Supplementary Table 5). The increased bacterial
richness was mainly attributed to the significantly increase detected in *Firmicutes* (ANOVA, $P = 0.011$),
*Actinobacteria* ($P = 0.002$), *Chloroflexi* ($P = 0.002$), *Deltaproteobacteria* ($P = 0.02$), and
*Alphaproteobacteria* ($P = 0.008$; Supplementary Table 6).
*3.3 The influence of permafrost thawing status and age on bacterial community structure*
Bray–Curtis distance based NMDS ordination plot revealed a clear separation of the bacterial
community structure by permafrost thawing status (Fig. 3a), while the separation by permafrost age
was less obvious (Fig. 3b). The results of DistLM analyses revealed that the measured soil factors,
thawing status, and age explained a total of 10.7 % of the bacterial community structure. TN was the
most important factor by explaining 7.2 % of the community structure ($P = 0.001$). This was followed
by C:N ratio, TOC, soil age and thawing status, which explained additional 3.5 % ($P = 0.028$), 3 % ($P$
$= 0.083$), 2.9 % ($P = 0.105$), and 2.8 % ($P = 0.111$), respectively.
PERMANOVA indicated that significantly different community structure was observed among the
various permafrost thawing status and ages (both $P < 0.001$, Supplementary Table 7), and an
interactive effect of the two existed ($P < 0.001$). Post–hoc analysis indicated that the community



structure differences were significantly different among the community structure in soils of different
permafrost thawing status (all $P < 0.1$, Supplementary Table 8). In contrast, significant differences were
only detected between the young– and older–aged permafrost soils (all $P < 0.05$, Supplementary Table
9), and between the medium– and ancient–aged soils ($P = 0.024$). PERMDISP analysis indicated that
the community homogeneity was not significantly different across the different permafrost thawing
status ($F(2, 42) = 0.193$, $P = 0.831$). Gradual transition of bacterial community structure due to
permafrost thawing was observed in each permafrost age category (Figs. 3c–f). Significantly different
soil bacterial community structure across the various thawing status was detected in the young, medium,
and ancient–aged permafrost (PERMANOVA, $P = 0.002$, $0.027$, and $0.016$, respectively,
Supplementary Table 10), but not in the old permafrost ($P = 0.124$). Similarly, significantly different
soil bacterial structure was also detected among the permafrost of different ages with the same thawing
status (Supplementary Table 11, Supplementary Fig. 2).
*3.4 The influence of permafrost thawing status and age on the community assembly of bacteria*
The average contribution of the stochasticity was 68 %, 74 %, and 86 % in the active, transition, and
frozen layers of the permafrost, and significant differences were detected between the active and frozen,
and between the transition and frozen layers (both $P < 0.05$, Supplementary Fig. 3a), but not between
the active and transition layers ($P = 0.15$). In contrast, the average contribution of stochasticity was
65 %, 76 %, 68 %, and 76 % for the young–, medium–, old–, and ancient–aged permafrost, with no
significant difference among the different aged permafrost being detected (all $P > 0.05$, Supplementary
Fig. 3b).
*3.5 Quantifying the influence of permafrost thawing status and age on bacterial richness and*
*community structure variation*
Structural equation modelling (SEM) revealed that both permafrost thawing status and age significantly
contributed to bacterial richness. Permafrost thawing status exhibited a higher contribution than age
(standard regression weight of 0.51 and 0.30, respectively, both $P < 0.05$) to bacterial richness (Fig. 4a).
In contrast, only permafrost thawing status exhibited a significant contribution to the NMDS1 of the
bacterial community structure (standard regression weight of 0.49, $P < 0.001$, Fig. 4b), while
permafrost age and thawing status both significantly contributed to the NMDS2 (standard regression





weight of 0.45 and 0.33, respectively, both $P < 0.01$).
**4 Discussion**
The bacteria richness was significantly higher in the active layer soil (Fig. 2a), and this is consistent
with the previous findings that permafrost thawing significantly increases bacterial richness in soil in
the Tibetan Plateau and the high Arctic (Chen et al., 2017; Schostag et al., 2019; Wu et al., 2018).
Permafrost thawing leads to an accelerated microbial degradation of soil organic carbon that can
generate a wide variety of metabolic products (Mueller et al., 2015). The increased metabolic product
diversity would lead an increased nutrient diversity and provide additional ecological niches for
bacteria (Hernández and Hobbie, 2010). This would explain the increased bacterial richness observed
in our study.
The soil bacteria in the young permafrost exhibited a stronger response to thawing than those in older
permafrost soils (Fig. 2b). The young permafrost soil demonstrated a higher relative abundance of
aliphatic carbon but lower carbohydrates than older permafrost soils (Mueller et al., 2015). Thus,
bacterial richness could be driven by carbon quality, but not quantity. It has been reported that the
degradation of complex carbon molecules requires extensive microbial collaboration, thus leads to a
more diverse microbial community in forest soil (Ding et al., 2015). Furthermore, an early study on the
freshwater ecosystem also confirmed that organic carbon composition determined bacterial richness
and community structure (Docherty et al., 2006). This is in agreement with the higher bacterial richness
detected in the active layer of the young permafrost soil (Fig. 2b).
The increased bacterial richness due to permafrost thawing was mainly attributed to *Firmicutes*,
*Actinobacteria*, *Chloroflexi*, *Deltaproteobacteria*, and *Alphaproteobacteria* in the young permafrost
soil (Supplementary Table 6). Increased transcriptional response of *Chlrofolexi* has been reported
during permafrost thawing (Coolen and Orsi, 2015), and may be attributed to their recalcitrate organic
matter degradation capacity (Colatriano et al., 2018). *Firmicutes* and *Actinobacteria* have been reported
to be more abundant in the frozen layer than in the active layer of permafrost soil due to their capacities
in maintaining metabolic activity and DNA repair mechanisms at low temperature (Johnson et al., 2007;
Tuorto et al., 2014). However, our results showed that their diversity may increase during permafrost
thawing, despite their reduction in relative abundance. *Alpha–* and *Delta–proteobacteria* were both



abundant in the upper permafrost soil in the Tibetan Plateau, and their relative abundance negatively
correlated with soil depth (Wu et al., 2017). *Alphaproteobacteria* was identified to be more abundant in
the active layer of the permafrost soil in Norway (Mueller et al., 2018). One possible explanation is that
the surface active layer may be the major location for root exudates, which favours
*Alphaprroteobacterai* (Morgalev et al., 2017). *Deltaproteobacteria* has been reported to have a strong
catabolic potential on the degradation of recalcitrate aromatic and other plant detritus (Jansson and Tas,
2014), thus also enhances their richness in the surface active layer of permafrost soil.
PERMANOVA and SEM both demonstrated statistically significant contributions of permafrost
thawing and age to soil bacterial community structure (Fig. 4b). However, bacterial communities were
better separated by thawing status than by age on NMDS plots (Figs. 3a and 3b). Furthermore, a
significantly higher contribution of determinism (lower stochasticity) was observed in the thawed
permafrost soils (active and transition layers) than in the permafrost layer, but not between the
permafrost soils with different ages (Supplementary Fig. 3). Collectively, this suggests that permafrost
thawing have a stronger influence on the bacterial community structure than permafrost age. Our
results is consistent with Mondav et al. (2017), who found that permafrost activity better separated the
community structure than soil depth in peatland permafrost soil in Sweden.
Permafrost thawing significantly increased determinism in bacterial community structure
(Supplementary Fig. 3). Increased determinism are frequently attributed to the enhanced environmental
filtering (Stegen et al., 2012). Our results demonstrated that nitrogen and the C:N ratio explained a
greater proportion of the bacterial community structure than TOC. This is consistent with the previous
findings that nitrogen availability strongly regulates microbial community structure and function in the
permafrost soils of Arctic and Tibetan Plateau (Chen et al., 2018; Chen et al., 2017; Yergeau et al.,
2010). Significantly different soil carbon and nitrogen were observed among the various permafrost
thawing statuses, but not among the different permafrost ages (Figs. 1a and 1c). Thus the changed
nutrients may explain the significant influence of thawing status on the community structure and
assembly processes. The community structure change due to permafrost thawing has also been
proposed to be due to the colonization of microorganisms in active layer (Monteux et al., 2018), which
coincides with the increased bacterial richness observed here (Fig. 2a).





The influence of permafrost age on bacterial community structure was weaker (Fig. 3b), with only
significantly different community structure being observed in the young– and medium–aged permafrost
soils (Supplementary Table 8). Substantial influence of permafrost age on community structure has
been reported previously (Mackelprang et al., 2017). Investigation on the podegenesis following
deglaciation also revealed distinct microbial community structure along the chronosequence (Freedman
and Zak, 2015). However, the community structure differences observed were much weaker than that
expected particularly between the old and ancient permafrost soils (Supplementary Table 9). This is
likely due to the strong influence of permafrost thawing, as thawing enhances environmental filtering
(Supplementary Fig. 3) and homogenizes community structure in soils with different ages. This is
confirmed by the significantly different bacterial community structure in permafrost soils of the same
age along the thawing gradient (except the old permafrost soil, Figs 3c–3f, Supplementary Table 10).
Our results also demonstrated that the soil community structure did not converge due to thawing
(Supplementary Fig. 2, Supplementary Table 11). This contradicts to previous studies (Deng et al.,
2015; Yuan et al., 2018) in the Arctic, but was consistent with Mackelprang (2011). The distinct
bacterial community structure in the various aged permafrost soils, yet under the same thawing status,
confirms the historic effects of permafrost age on the community structure during permafrost thawing.
The distinct bacterial community structure is likely to result in different metabolic functions (Brown
and Forsythe, 1974), thus the significantly different bacterial structure under the same thawing status
may lead to different organic carbon degradation capacities. Furthermore, older permafrosts enriches
pathways involved in the degradation of recalcitrant biomass, while decreases pathways associated
with starch and sucrose metabolism comparing with younger soils (Mackelprang et al., 2017). Thus, the
thawing of permafrost soils of different ages may also lead to distinct soil carbon degradation schemes.
**5 Conclusion**
Our results demonstrated that permafrost thawing consistently exhibited greater influence on bacterial
richness and community structure than permafrost age. However, permafrost age alters the response of
permafrost soil bacteria to thawing, with a stronger response to thawing observed in the young than
older permafrost soils. The different community structure during permafrost thawing may present
distinct metabolic potentials for soil organic carbon cycling, and may ultimately alter the carbon
emission scheme.



**Data availability**

Sequence data generated in the present study have been deposited to the National Center for
Biotechnology Information (NCBI) Sequence Read Archive under the ID PRJNA554442.
**Author contributions**
WK conceived the study and developed the idea with MJ, TZ and HZ performed DNA extraction, MJ
conducted the data statistical analysis. MJ and WK wrote the first draft of the manuscript, CL and XD
revised the manuscript substantially. All authors read and approved the final manuscript.
**Competing interests**
The authors declare that they have no conflict of interest.
**Acknowledgements**
This project was financially supported by Chinese Academy of Sciences [grant numbers
XDA19070304, QYZDB-SSW-DQC033 and XDA20050101], and National Natural Science
Foundation of China [grant number 41771303]. We greatly thank Dr. J Kao−Kniffin for kindly
providing the permafrost soil samples for this study.





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



**Figure legends**
Fig. 1 Total organic carbon (a and b) and total nitrogen (c and d) with the permafrost age (young,
medium, old, and ancient) and permafrost thawing status (active, transition and permanently frozen).
Fig. 2 Bacterial richness with the permafrost thawing status (a) and age (b). The richness is indicated
byoperational taxonomic unit (OTU) number. Different letters indicate significant difference at $P < 0.05$.
Young, medium, old, and ancient are permafrost soil ages, active, transition, and permanently frozen
are permafrost thawing status.

Fig. 3 NMDS plots showing the bacterial community structure of different thawing status (a) and
permafrost age (b). The bacterial community structure of different thawing status in the young, medium,
old, and ancient permafrost soils are shown in (c)–(f). Active, transition, and permanently frozen are
permafrost thawing status.

Fig. 4 The relative importance of permafrost thawing status and age on bacterial richness (a) and
community structure (b) based on structural equation modelling. The community structure variation
was assessed by the 1st and 2nd axis coordinates of the NMDS plot (NMDS1 and NMDS2). Numbers
adjacent to arrows are the absolute value of the path coefficients, indicative of the standardized effect
size of the relationship.*: $P < 0.05$, **: $P < 0.01$ and ***: $P < 0.001$. The arrow thickness represents the
strength of the relationship.





Fig. 1

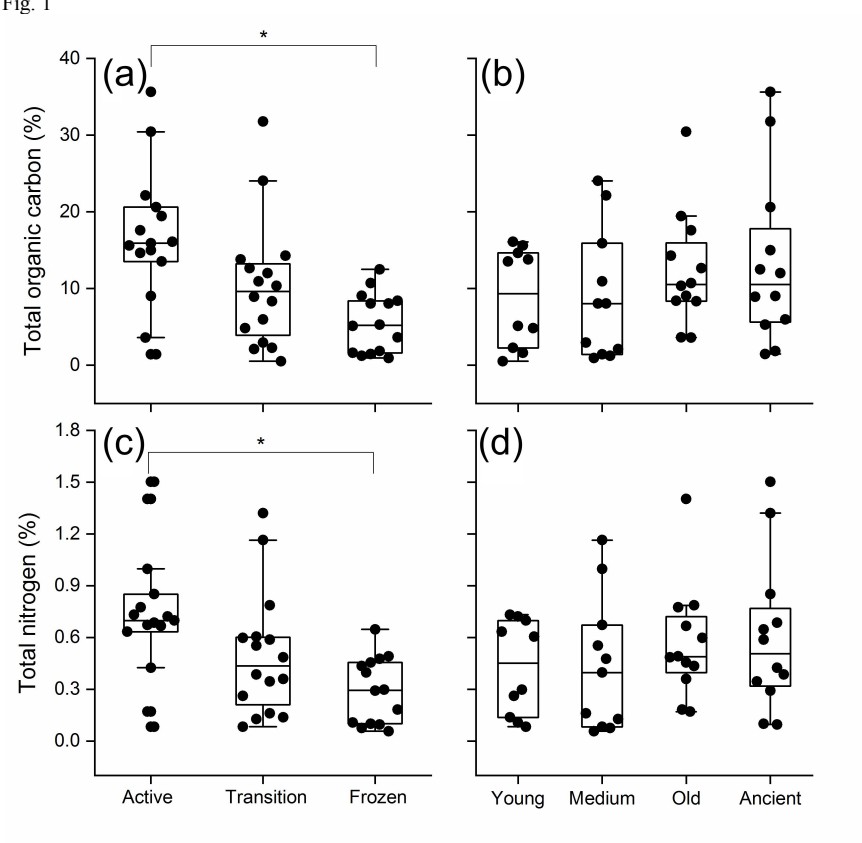





Fig. 2

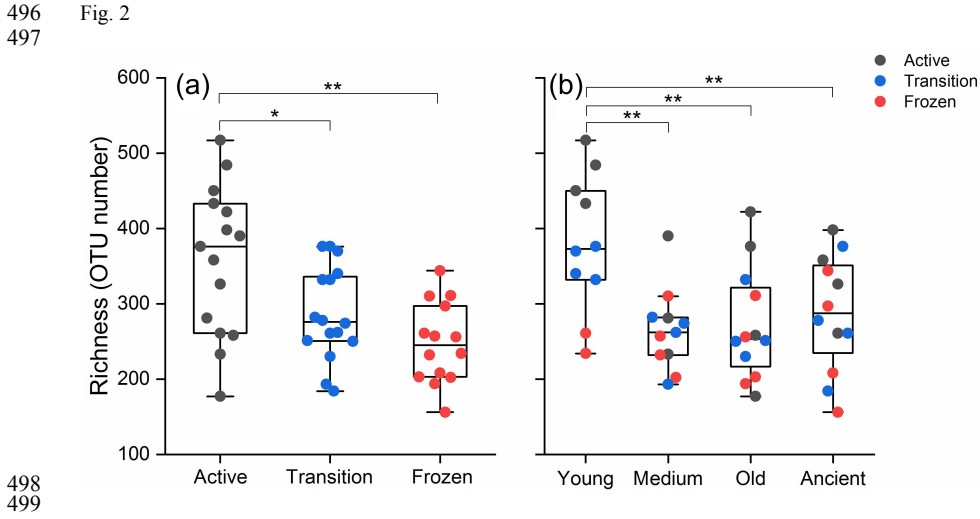






Fig. 3

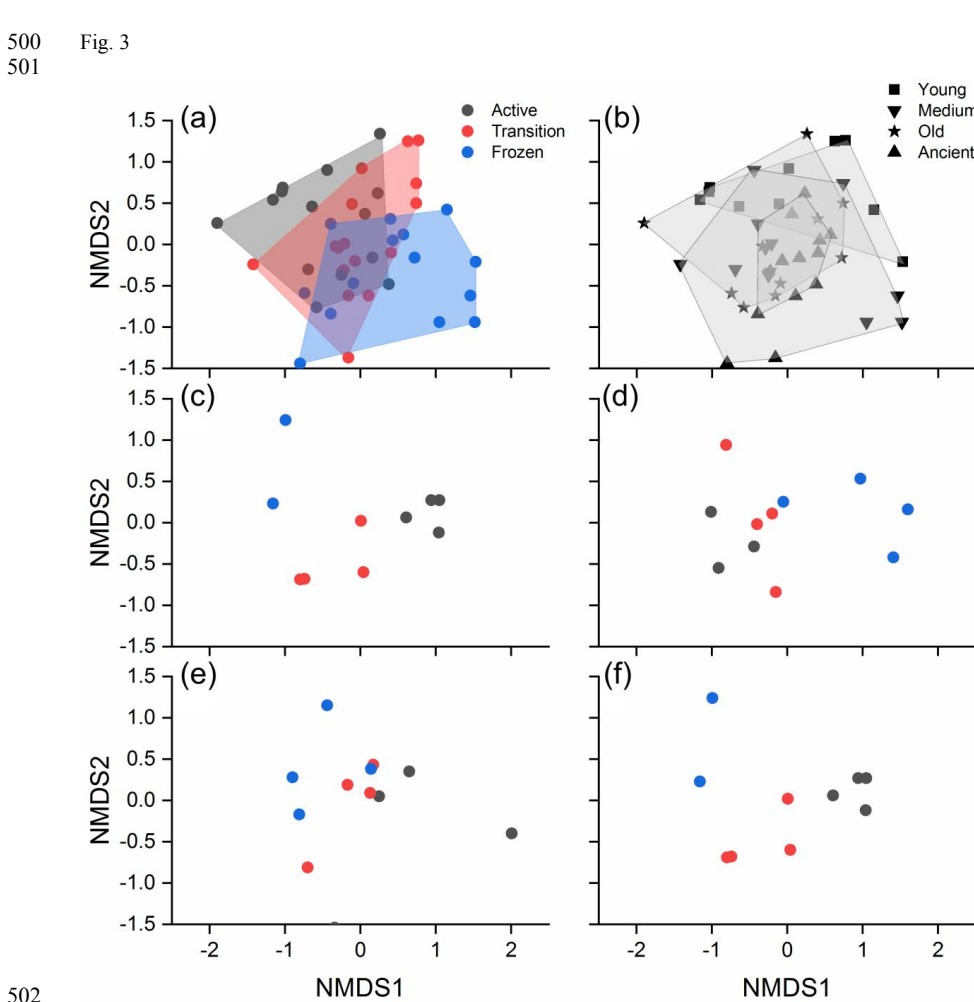






Fig. 4

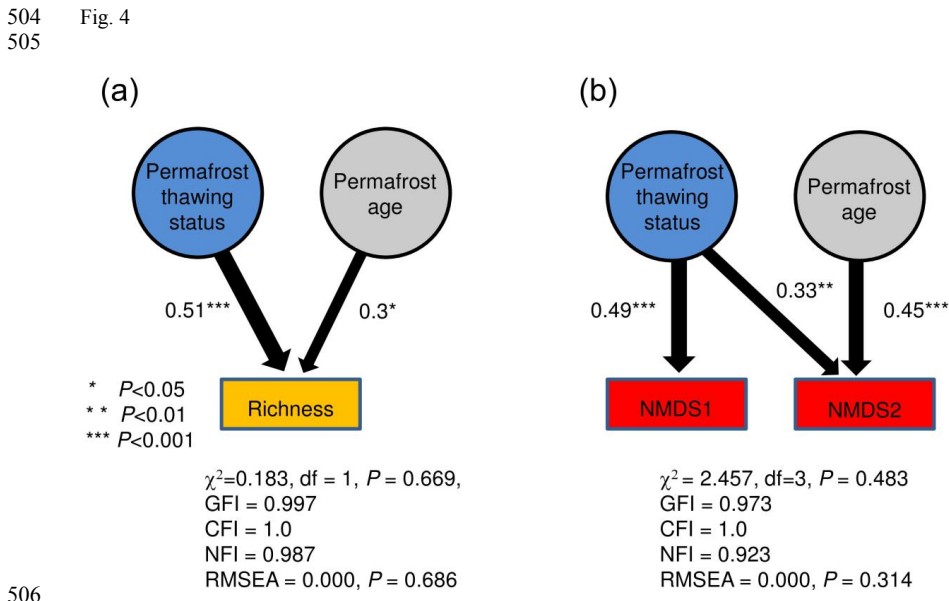
