# Peer review of "Permafrost thawing exhibits a greater influence on bacterial richness and community structure than permafrost age in Arctic permafrost soils"

_The Cryosphere, 2020_

## Referee Comment (RC1) · Anonymous Referee #1 · 14 Apr 2020

This is a technically correct manuscript on a currently relevant topic in the context of climate change and biogeochemical cycles - the response of microbes to permafrost thawing. The study shows changes in bacterial community structure and richness of drained lake basins with permafrost soil age and permafrost thawing status (active, transition and permanently frozen soil layer). In addition there is data on soil carbon and nitrogen. The results are presented clearly and the figures are well prepared.

Major concerns:

1. Are the samples in this study from the same soil cores as those in Kao-Kniffin et al. 2015, which is cited in the section on sampling? Kao-Kniffin et al. 2015 also describe

bacterial communities with permafrost soil age and thawing status. If the soil cores are the same, please make it clear in the aims why a second analysis of bacterial communities in these samples is needed and explain what new this study adds. In any case, please take the results of Kao-Kniffin et al. 2015 into account in the discussion, especially as their conclusion (communities in active layers converge) seems to be the opposite from this manuscript (no convergence of active layer communities).

2. I am concerned that the connection of soil layers to thawing status is too simplified and does not take into account variation in the soil profile. Was the soil structure/chemical composition of the profiles homogeneous with depth? The description of organic layer on l. 94-96, Fig. 1 and Kao-Kniffin et al. 2015 and Mueller et al. 2015 cited in the manuscript suggest they were not. In this case, the differences in bacterial communities between soil layers cannot be directly interpreted as a thawing response (l.37., l.318), because the state of the system before thawing is not known and the differences between the layers can be due differences in other soil properties (for example organic vs. mineral layer). It is possible to compare the active, transition and frozen layers with permafrost age but that seems to have already been done by Kao-Kniffin et al. 2015? In any case, the issue of other differences between the soil layers than thawing status should be better taken into account in the manuscript. Do soil carbon and nitrogen explain the community changes?

Minor comments:

l. 82-91 Please indicate where your replicate samples come from and how many there are. Here it is mentioned that there are four age classes and one soil core per age class, but the figures show a lot more data points (over 40?).

Table S6: I am confused how mean relative abundance can be over 100%. Could you clarify? Please also check the definition of SD (should be standard deviation?).

Minor comments on spelling and grammar:

l. 30 Deltaproteobacterai -> Deltaproteobacteria

l. 95 vary -> varies

l. 253 early -> earlier

l. 270 Alphaprroteobacterai -> Alphaproteobacteria

l. 272 Please check language. What enhances their richness?

l. 292 have -> has

l. 280 results is -> results are

---

## Referee Comment (RC2) · Anonymous Referee #2 · 19 Jul 2020

This manuscript reports significant influences of both permafrost thawing and age on bacterial richness and community structure. It also documented that permafrost thawing increased the contribution of determinism to bacterial community assembly, but didn't lead to community convergence. The study then showed that permafrost thawing had a greater influence on bacterial community than permafrost age. They extrapolate their findings to highlight that permafrost thawing in different ages can lead to distinct bacterial community compositions and different soil organic carbon degradation processes. The manuscript is well organized and figures are well prepared. I have several major concerns about this manuscript: 1. I am not a expert in permafrost. But I noticed that the samples used in this study should be the same samples reported in Kao-Kniffin

et al. 2015. In this study, the permafrost age of different samples had been measured by Kao-Kniffin et al. 2015. However, basin age is used in that study instead of permafrost age. Does it mean that basin age is equal to permafrost age? If so, Why did this study reported different soil total organic carbon and total nitrogen from Kao-Kniffin et al. 2015? 2. It seems that Kao-Kniffin et al. 2015 also used amplicon sequencing of 16S rRNA gene to analyze bacterial communities in different permafrost ages and thawing status. They found that community composition appeared to converge in the active layer, however, the authors in this study didn't observe the community convergence due to permafrost thawing. Can you explain why you reanalyzed bacterial communities of these samples? At least, please compare your study with the results of Kao-Kniffin et al. 2015 and provide more discussion. 3. The thickness of active, transition and permafrost layers should be different in young, medium, old and ancient permafrost. Please provide more information about the soil profile of different layers in four kinds of permafrost. More variables should be taken into account to undermining the mechanism of bacterial response to permafrost thawing in different permafrost age. I'm not sure that structural equation modelling is a good method to quantify the relative importance of permafrost thawing status and age on bacterial community without any other environmental variables. Please incorporate more variables in structural equation modelling to show how permafrost thawing status and age influenced bacterial community directly or indirectly Minor comments: l. Fig.2 Can you provide information about bacterial phylogenetic diversity of bacteria in different permafrost age and thawing status? 2. Fig.4 the path coefficients in structural equation modelling can indicate the positive or negative correlations between two variables. Therefore, the raw value should be shown here, instead of the absolute value. 3. Line 95 vary -> varies 4. Line 272 Please rewrite this sentence. 5. 280 results is -> results are

---

## Short Comment (SC1) · 5 Aug 2020

The authors would like to thank the reviewer for the constructive feedback, and the thorough assessment of the manuscript. Below we provide a point-to-point response to each comment, reviewer comments are given in black, responses are given in blue. Additionally, we have included details of how we intend to address these changes in a revised submission.

This manuscript reports significant influences of both permafrost thawing and age on bacterial richness and community structure. It also documented that permafrost thawing increased the contribution of determinism to bacterial community assembly,

but didn't lead to community convergence. The study then showed that permafrost thawing had a greater influence on bacterial community than permafrost age. They extrapolate their findings to highlight that permafrost thawing in different ages can lead to distinct bacterial community compositions and different soil organic carbon degradation processes. The manuscript is well organized and figures are well prepared. I have several major concerns about this manuscript:

Q1. I am not an expert in permafrost. But I noticed that the samples used in this study should be the same samples reported in Kao-Kniffinet al. 2015. In this study, the permafrost age of different samples had been measured by Kao-Kniffin et al. 2015. However, basin age is used in that study instead of permafrost age. Does it mean that basin age is equal to permafrost age? If so, Why did this study reported different soil total organic carbon and total nitrogen from Kao-Kniffin et al. 2015?

Response:
The "basin age" used by Kao-Kniffin et al. 2015 was obtained from Hinkel et al., 2003. This age is calculated using radiocarbon dating on sample at the interface of the lacustrine sediment and the in situ peat, which represents the point in time of lake drainage and revegetation of the basin surface. The basin drainage is associated with vegetation establishment, organic matter accumulation, and ice-wedge growth below the drained lake basin (Hinkel et al., 2003). Therefore, we used the basin age as a proxy for the formation of permafrost (permafrost age), and explored the influence of permafrost age on the response of bacteria to permafrost thawing. We did not use the term basin age, as we feel that permafrost age is more meaningful than basin age in this case. We apologize for not stating this clearly, and following sentences will be added to clarify this.

Amended manuscript:
Approximately 20% of the Arctic coastal plains of northern Alaska contain thaw lakes drained at various stages since the mid-Holocene, which were then developed into ice-rich permafrost (Hinkel et al., 2003). These drained lake basins contain soils ranging from freshly developed organic layers on sediments to fully developed ancient permafrost soils (Mueller et al., 2015). By using the drained thaw lake basin age as a proxy for the time of permafrost formation, it provides an opportunity to investigate the influence of permafrost age on the microbial community during permafrost degradation.

Regarding the inconsistent values of environment factors with Kao-Kniffinet al. 2015, these environmental factors (SOC and TN) were independently measured in the present study. They are not substantially different from those reported in the Kao-Kniffinet al. 2015 (please see Supplement table 1 attached), and the differences could be related to sample storage and the equipment used.

Q2. It seems that Kao-Kniffin et al. 2015 also used amplicon sequencing of 16S rRNA gene to analyze bacterial communities in different permafrost ages and thawing status. They found that community composition appeared to converge in the active layer, however, the authors in this study didn't observe the community convergence due to permafrost thawing. Can you explain why you reanalyzed bacterial communities of these samples? At least, please compare your study with the results of Kao-Kniffin et al. 2015 and provide more discussion.

Response:
We appreciate the reviewer for this comment. In this manuscript, we focused on the bacterial community only, whereas Kao-Kniffin et al. 2015 investigated the community structure of the entire prokaryotes (Both bacteria and archaea). For bacteria, Kao-Kniffin presented the taxonomic composition, community phylogenetic distance, and biomass. Thus, the interactive influence of permafrost age and thawing on bacterial

diversity, community structure, and assembly processes still remain unexplored. Therefore, we believe further investigation is necessary and could provide essential knowledge on how permafrost age interact with permafrost thawing.

To address the reviewer's comments, we will add the following sentences to clarify our aims and distinguish our work from Kao–Kniffin et al.:

An earlier study has revealed a high abundance of Candidatus Methanoflorens archaeon in the community (Kao–Kniffin et al., 2015), but how the bacteria in the permafrost of various ages would respond to thawing remains undiscussed. Thus, we take this opportunity to re-analyze these samples to investigate the interactive influence of permafrost thawing and age on the permafrost soil bacterial community.

Following sentences will be added to discuss the inconsistency between our work and Kao–Kniffin et al on community convergence:

Our results demonstrated that the bacterial community structure did not converge due to permafrost thawing, as reflected by the non-significant difference in sample heterogeneity among the various permafrost layers (Supplementary Fig. 3, Supplementary Table 11). This contradicts previous studies (Deng et al., 2015; Yuan et al., 2018) in the Arctic, but was consistent with Mackelprang (2011). Our results also contradict to Kao–Kniffin et al. (2015), which reported a lower prokaryotic community differences in the active layer than in the transition and permanently frozen permafrost. Several reasons could cause this inconsistency. Firstly, different microbial communities were targeted. Kao–Kniffin et al. (2015) focused on the Archaeal community, and a single archaeon OTU accounted for over 30% of the community (Fig. 3 in Kao-Kniffin et al., 2015). This may drive the convergence of the prokaryotic community. In comparison, only bacterial community were targeted in the present study, and an early study has reveal distinct community structure of bacteria and archaea with archeal demonstrating lower variation across soil depth (Frank-Fahle et al., 2014). Furthremore,
the inconsistency may also related to the different community dissimilarity metrics used. Kao–Kniffin et al. (2015) used unweighted UniFrac, which only account for the phylogenetic closeness of the OTUs, and the relative abundance was not considered. This is distinctively different from the Bray-Curtis dissimilarity used in this study, and it has been reported that unweighted and weighted community metrics examine different features of the community (Lozupone et al., 2011).

Q3. The thickness of active, transition and permafrost layers should be different in young, medium, old and ancient permafrost. Please provide more information about the soil profile of different layers in four kinds of permafrost. More variables should be taken into account to undermining the mechanism of bacterial response to permafrost thawing in different permafrost age. I'm not sure that structural equation modelling is a good method to quantify the relative importance of permafrost thawing status and age on bacterial community without any other environmental variables. Please incorporate more variables in structural equation modelling to show how permafrost thawing status and age influenced bacterial community directly or indirectly.

Response:
We appreciate the reviewer for this comment. We agree with reviewer that soil profile characteristics are important in determining the response of bacteria to permafrost thawing. Unfortunately, we have only obtained a small quantity of soil samples, and this does not allow us to measure soil properties in great detail. Therefore, we decided to focus on TOC and TN, which are known to be substantially different in the permafrost of different ages and thawing status (Mueller et al., 2015). We will amend the manuscript to acknowledge the limitation of our work and propose further works that are required to identify the environmental factors that shape bacterial response to permafrost thawing.

Amended manuscript (method section)
Due to sample quantity limitation, only two of the most important soil physicochemical properties: total organic carbon (TOC) and total nitrogen (TN) were measured using dry combustion (Vario MAX CNS Analyzer, Elementar, Hanau, Germany). These factors have been reported to substantially vary in samles with different permafrost age and thawing status (Mueller et al., 2015). For other soil properties and soil profile descriptions of the different layers please see Kao-Kniffin et al (2005).

To address the reviewer's concern and emphasize the importance of environmental heterogeneity in the different permafrost layers, we will add the following paragraph to the manuscript:

Amended manuscript (discussion section)
Bacterial community structure in the active layer is more similar to the transition layer than to the permanently frozen layer (Fig. 3). This is consistent with those observed in other Arctic permafrost (Monteux et al, 2018, Deng et al., 2015), confirming that thawing can homogenize bacterial community structure of different soil depths. However, significant differences in the bacterial community were still observed between the active and transition layers (Supplementary Table 8), instead of being identical (Monteux et al, 2018). This could be due to physiochemical heterogeneity between the soils of different permafrost layers (Fig. 1, Kao-Kniffin, et al., 2015, Mueller et al. 2015). Thus, physicochemical properties (such as the total nitrogen observed here) differences also contribute to the bacterial community heterogeneity and led to the significantly different bacterial communities observed.

Amended manuscript (Conclusion section)
Further studies are required to identify the environmental and historical factors that lead to the distinct response of bacteria in the permafrost of different ages.

Using ranked soil age in SEM to investigate the influence of pedogenic development has been used by Laliberté et al., 2017. Here, we use this idea to investigate the influence of permafrost age on the response of bacteria to permafrost thawing. We repeated the SEM analysis incorporating the TOC and TN as suggested by the reviewer, and a large proportion of the richness and community structure were still explained by the permafrost age and thawing status. Therefore, the permafrost age and thawing status can influence richness and community structure via other unmeasured environmental factors or historical effects. The present study does not attempt to identify these factors, but instead, we tried to raise the awareness that different aged permafrost may response to climate change induced permafrost thawing differently. To address this comment, we have incorporated TOC and TN into the SEM (attached as supplementary Fig 1), performed Random Forest analysis to identify the importance of permafrost age and thawing status (attached as supplementary Fig 2), and also amended the results and conclusion section to address the requirement of identifying environmental/historical factors that directly cause the different responses of bacteria in the permafrost of various ages.

Amended manuscript (Results section)
Structural equation modelling (SEM) revealed that both permafrost thawing status and age significantly contributed to bacterial richness. Permafrost thawing status exhibited a higher contribution than age (standard regression weight of 0.51 and -0.30, respectively, both P < 0.05) to bacterial richness (Fig. 4a). However, the influence of TOC and TN on bacterial richness was not observed. This is consistent with the Random Forest analysis results, which only identified the permafrost thawing and age as the significant determinants of bacterial richness (Supplementary Fig. 5). For community structure, permafrost thawing exhibited an indirect influence on NMDS1 via TN (standard regression weight of 0.58 and -0.63, both P < 0.001, Fig. 4b). In comparison, both permafrost age and thawing status significantly contributed to the NMDS2 (standard regression weight of -0.34 and 0.59, respectively, both P < 0.01),

while TN also exhibited an significant influence on NMDS2 (-0.49, P = 0.002). The significant contribution of TN, permafrost thawing and age were consistently identified using Random Forest approach (Supplementary Fig. 5).

Amended manuscript (Conclusion section)
Further studies are required to identify the environmental and historical factors that lead to the distinct response of bacterial in the permafrost of different ages.

Minor comments:
Q4. Fig.2 Can you provide information about bacterial phylogenetic diversity of bacteria in different permafrost age and thawing status?

Response:

We appreciate the reviewer for this comment. The results of Phylogenetic diversity and Shannon diversity will be added to the results and discussion sections.

Amended manuscript (Results section)

In comparison, the influence of permafrost age and thawing on bacteria Shannon diversity were non-significant (Two-way ANOVA, P = 0.058 and 0.53, respectively, Supplementary Fig. 2). This contrastively different from those observed on phylogenetic diversity, where significant influence was observed for thawing (P = 0.015 and 0.001, respectively).

Amended manuscript (Discussion section)

Furthermore, the phylogenetic diversity exhibited greater sensitivity to permafrost thawing than to the Shannon diversity. As phylogenetically close-related microorganisms have similar habitat associations, phylogeny-based community metrics could infer potential community functional change (Stegen et al., 2012). Hence, this suggests that community function could be more sensitive to permafrost thawing than the community composition.

Q5. Fig.4 the path coefficients in structural equation modelling can indicate the positive or negative correlations between two variables. Therefore, the raw value should be shown here, instead of the absolute value.

Response:
We appreciate the reviewer for this comment. We will includ the signs of path coefficient in the figure (attached as supplment figure 1) and amend the manuscript accordingly.

Amended manuscript
Structural equation modelling (SEM) revealed that both permafrost thawing status and age significantly contributed to bacterial richness. Permafrost thawing status exhibited a higher contribution than age (standard regression weight of 0.51 and -0.30, respectively, both $P < 0.05$) to bacterial richness (Fig. 4a). However, the influence of TOC and TN on bacterial richness was not observed. This is consistent with the Random Forest analysis results, which only identified the permafrost thawing and age as the significant determinants of bacterial richness (Supplementary Fig. 5). For community structure, permafrost thawing exhibited an indirect influence on NMDS1 via TN (standard regression weight of 0.58 and -0.63, both $P < 0.001$, Fig. 4b). In comparison, both permafrost age and thawing status significantly contributed to the NMDS2 (standard regression weight of -0.34 and 0.59, respectively, both $P < 0.01$), while TN also exhibited an significant influence on NMDS2 (-0.49, $P = 0.002$). The significant contribution of TN, permafrost thawing and age were consistently identified

using Random Forest approach (Supplementary Fig. 5).

Q6. Line 95 vary -> varies

The mis-spelling is now corrected, and the amended manuscript is:

The surface organic layer thickness varies with permafrost age, which was < 5, 10–15, 15–30, and 40–50 cm for the young, medium, old, and ancient–aged permafrost soils (Kao–Kniffin et al., 2015)

Q7. Line272 Please rewrite this sentence.

The sentence is rephrased as:

Deltaproteobacteria has strong catabolic potentials to decompose recalcitrate aromatic compounds and other plant detritus (Jansson and Tas, 2014), which may explain their enhanced richness in the surface active layer of permafrost soil.

Q8. 280 results is -> results are

The mis-spelling is now corrected, and the amended manuscript is:

Our results are consistent with Mondav et al.(2017), who found that permafrost activity better separated the community structure than soil depth in peatland permafrost soil in Sweden.
Please also note the supplement to this comment:
https://tc.copernicus.org/preprints/tc-2020-39/tc-2020-39-SC1-supplement.pdf
* * *
Table 1 Comparison of soil organic carbon (SOC) and total nitrogen (TN) using in the present study and Kao-Kniffin et al.

| | SOC (%) | | TN (%) | |
|---|---|---|---|---|
| | The present study | Kao-Kniffin et al. 2015. | The present study | Kao-Kniffin et al. 2015. |
| Basin age | | | | |
| Young | 8.8 (2.0) | 6.9 (1.9) | 0.43 (0.09) | 0.43 (0.11) |
| Medium | 9.8 (2.5) | 8.4 (1.6) | 0.45 (0.11) | 0.49 (0.09) |
| Old | 12.4 (2.2) | 13.4 (1.6) | 0.57 (0.10) | 0.77 (0.09) |
| Ancient | 13.3 (3.2) | 10.7 (2.5) | 0.60 (0.13) | 0.62 (0.14) |
| Soil depth layer | | | | |
| Active | 16.9 (2.1) | 14.1 (2.1) | 0.73 (0.09) | 0.78 (0.11) |
| Transition | 10.4 (2.1) | 9.6 (1.8) | 0.50 (0.09) | 0.57 (0.11) |
| Permafrost | 5.6 (1.0) | 5.8 (1.4) | 0.29 (0.05) | 0.36 (0.09) |

The values indicate means and standard error (in parentheses)

**Fig. 1.** Comparison of TOC and TN between the present study and Kao-Kniffin et al. 2015

[Figure]

Fig. 2. The relative importance of permafrost thawing status and age on bacterial richness (a)
and community structure (b)based on structural equation modelling

a

TN

TOC

Permafrost age

Thawing status

b

TN

TOC

Permafrost age

Thawing status

c

TN

TOC

Permafrost age

Thawing status

0    2    4    6    8    10   12   14   16
Percentage increase in Mean Square Error (MSE)

**Fig. 3.** Random Forest analysis results showing the contribution of permafrost age, thawing status, TOC and TN on the bacterial richness (a), nmds1 (b), and nmds2 (c))

---

## Short Comment (SC2) · 5 Aug 2020

The authors would like to thank the reviewer for the constructive feedback, and the thorough assessment of the manuscript. Below we provide a point-to-point response to each comment, reviewer comments are given in black, responses are given in blue. Additionally, we have included details of how we intend to address these changes in a revised submission.

This is a technically correct manuscript on a currently relevant topic in the context of climate change and biogeochemical cycles - the response of microbes to permafrost thawing. The study shows changes in bacterial community structure and richness of

drained lake basins with permafrost soil age and permafrost thawing status (active, transition and permanently frozen soil layer). In addition, there is data on soil carbon and nitrogen. The results are presented clearly and the figures are well prepared.

Major concerns:
Q1. Are the samples in this study from the same soil cores as those in Kao-Kniffin et al. 2015, which is cited in the section on sampling? Kao-Kniffin et al. 2015 also describe bacterial communities with permafrost soil age and thawing status. If the soil cores are the same, please make it clear in the aims why a second analysis of bacterial communities in these samples is needed and explain what new this study adds. In any case, please take the results of Kao-Kniffin et al. 2015 into account in the discussion, especially as their conclusion (communities in active layers converge) seems to be the opposite from this manuscript (no convergence of active layer communities).

Response:

We appreciate the reviewer for this comment. In this manuscript, we focused on the bacterial community only, whereas Kao-Kniffin et al. 2015 investigated the community structure of the entire prokaryotes (Both bacteria and archaea). For bacteria, Kao-Kniffin presented the taxonomic composition, community phylogenetic distance, and biomass. Thus, the interactive influence of permafrost age and thawing on bacterial diversity, community structure, and assembly processes still remain unexplored. Therefore, we believe further investigation is necessary and could provide essential knowledge on how permafrost age would influence the bacterial community interactively with permafrost thawing. To address the reviewer's comments, we will add the following sentences to clarify our aims and distinguish our work from Kao–Kniffin et al.:

An earlier study has revealed a high abundance Candidatus Methanoflorens archaeon in the community (Kao–Kniffin et al., 2015), but how the bacteria in the permafrost of various ages would respond to thawing remains undiscussed. Thus, we take this opportunity to re-analyze these samples to investigate the interactive influence of permafrost thawing and age on the permafrost soil bacterial community.

Following sentences are to be added to discuss the inconsistency between our work and Kao–Kniffin et al on community convergence:

Our results demonstrated that the bacterial community structure did not converge due to permafrost thawing, as reflected by the non-significant difference in sample heterogeneity among the various permafrost layers (Supplementary Fig. 3, Supplementary Table 11). This contradicts previous studies (Deng et al., 2015; Yuan et al., 2018) in the Arctic, but was consistent with Mackelprang (2011). Our results also contradict to Kao–Kniffin et al. (2015), which reported a lower prokaryotic community differences in the active layer than in the transition and permanently frozen permafrost. Several reasons could cause this inconsistency. Firstly, different microbial communities were targeted. Kao–Kniffin et al. (2015) focused on the Archaeal community, and a single archaeon OTU accounted for over 30% of the community (Fig. 3 in Kao-Kniffin et al., 2015). This may drive the convergence of the prokaryotic community. In comparison, only bacterial community were targeted in the present study, and an early study has reveal distinct community structure of bacteria and archaea with archeal demonstrating lower variation across soil depth (Frank-Fahle et al., 2014). Furthremore, the inconsistency may also related to the different community dissimilarity metrics used. Kao–Kniffin et al. (2015) used unweighted UniFrac, which only account for the phylogenetic closeness of the OTUs, and the relative abundance was not considered. This is distinctively different from the Bray-Curtis dissimilarity used in this study, and it has been reported that unweighted and weighted community metrics examine different

features of the community (Lozupone et al., 2011).

Q2. I am concerned that the connection of soil layers to thawing status is too simplified and does not take into account variation in the soil profile. Was the soil structure/chemical composition of the profiles homogeneous with depth? The description of organic layer on l. 94-96, Fig. 1 and Kao-Kniffin et al. 2015 and Mueller et al. 2015 cited in the manuscript suggest they were not. In this case, the differences in bacterial communities between soil layers cannot be directly interpreted as a thawing response (l.37., l.318), because the state of the system before thawing is not known and the differences between the layers can be due differences in other soil properties (for example organic vs. mineral layer). It is possible to compare the active, transition and frozen layers with permafrost age but that seems to have already been done by Kao-Kniffin et al. 2015? In any case, the issue of other differences between the soil layers than thawing status should be better taken into account in the manuscript. Do soil carbon and nitrogen explain the community changes?

We appreciate the reviewer to raise this concern. As the reviewer has pointed out, the physicochemical properties of the soils in different permafrost depths are not homogenous. Hence, the response of microbial community to thawing could be the collective effects of both thawing and the environmental factors difference. However, microbial transformation (as a result of permafrost thawing) would substantially change the quantity and composition of organic compounds (Mueller et al. 2015). Thus, soil physicochemical properties and bacterial community structure are interactive, and we have to admit that the individual influence would be very difficult to disentangle. Nevertheless, Mondav et al. (2017) reported that permafrost thawing has a stronger influence on microbial community structure than soil depth. To address the reviewer's concern and emphasize the importance of environmental heterogeneity in the different permafrost layers, we will add the following paragraph to the manuscript:

Bacterial community structure in the active layer is more similar to the transition layer than to the permanently frozen layer (Fig. 3). This is consistent with those observed in other Arctic permafrost (Monteux et al, 2018, Deng et al., 2015), confirming that thawing can homogenize bacterial community structureof different soil depths. However, significantdifferences in the bacterial community were still observed between the active and transition layers (Supplementary Table 8), instead of being identical (Monteux et al, 2018). This could be due to physiochemical heterogeneitybetween the soils in different permafrost layers (Fig. 1, Kao-Kniffin, et al., 2015, Mueller et al. 2015). Thus, other unmeasured physicochemical properties (such as the total nitrogen) in the different permafrost layers also contributed to the bacterial community heterogeneity and led to the significantly different bacterialcommunities.

We will also add the following sentence to the conclusion section, which demands further investigation to identify the factors (both environmental and historical) that caused the distinct microbial response to thawing.

Further studies are required to identify the environmental and historical factors that lead to the distinct response of bacterial in the permafrost of different ages.

Minor comments:

Q3. 82-91 Please indicate where your replicate samples come from and how many there are. Here it is mentioned that there are four age classes and one soil core per age class, but the figures show a lot more data points (over 40?). Table S6: I am confused how mean relative abundance can be over 100

We thank the reviewer for this comment, the following sentence will be added to clarify

the number of replicates used.

In brief, 16 soil cores were collected along a chronosequence of drained lake basin, spanning in age from young (< 50 yr old), medium (< 300 yr old), old (< 3,000 yr old), to ancient (3,000–5,000 yr old) in April 2010.
And
For each permafrost age-layer combination, there were four sample replicates, except for the young frozen permafrost, which had only two.

For Table S6, we apologize for the mistake. The number presented is the bacterial richness (i.e., number of OTUs observed), but not a percentage number, the amended table is attached in the supplementary file. The spelling of S.D. is corrected throughout the manuscript.

Minor comments on spelling and grammar:

Q4. The spelling and grammar mistakes have been corrected as indicated by the reviewer.

l. 30 Deltaproteobacterai -> Deltaproteobacteria

The mis-spelling is now corrected, and the amended manuscript is:

The bacterial richness was significantly higher in the young and thawed permafrost, and the richness increase was mainly observed in Firmicutes, Actinobacteria, Chloroflexi, Deltaproteobacteria, and Alphaproteobacteria.

l. 95 vary -> varies

The mis-spelling is now corrected, and the amended manuscript is:

The surface organic layer thickness varies with permafrost age, which was < 5, 10–15, 15–30, and 40–50 cm for the young, medium, old, and ancient–aged permafrost soils (Kao–Kniffin et al., 2015)

l. 253 early -> earlier

The mis-spelling is now corrected, and the amended manuscript is:

Furthermore, an earlier study on the freshwater ecosystem also confirmed that organic carbon composition determined bacterial richness and community structure

l. 270 Alphaprroteobacterai -> Alphaproteobacteria

The mis-spelling is now corrected, and the amended manuscript is:

One possible explanation is that the surface active layer may be the major location for root exudates, which favours Alphaproteobacteria

l. 272 Please check language. What enhances their richness?

The sentence is now rephrased as:

Deltaproteobacteria has been reported to have a strong catabolic potential on the degradation of recalcitrate aromatic and other plant detritus (Jansson and Tas, 2014), which enhances their richness in the surface active layer of permafrost soil.

l. 292 have -> has

The mis-spelling is now corrected, and the amended manuscript is:

Collectively, this suggests that permafrost thawing has a stronger influence on the bacterial community structure than permafrost age.

l. 280 results is -> results are

The mis-spelling is now corrected, and the amended manuscript is:

Our results are consistent with Mondav et al.(2017), who found that permafrost activity better separated the community structure than soil depth in peatland permafrost soil in Sweden.

|  |  | Mean±S.D. | Active | Transition | Frozen |
|---|---|---|---|---|---|
| *Firmicutes* | Active | 87±15 | - | - | - |
|  | Transition | 47±15 | **0.013** | - | - |
|  | Frozen | 49±8 | **0.042** | 0.986 | - |
| *Actinobacteria* | Active | 128±14 | - | - | - |
|  | Transition | 106±10 | 0.062 | **-** | - |
|  | Frozen | 71±2 | **0.002** | **0.021** | - |
| *Chloroflexi* | Active | 36±4 | - | - | - |
|  | Transition | 29±3 | 0.095 | - | - |
|  | Frozen | 16±6 | **0.002** | **0.02** | - |
| *Alphaproteobacteria* | Active | 35±6 | - | - | - |
|  | Transition | 19±7 | 0.016 | - | - |
|  | Frozen | 16±4 | 0.016 | 0.774 | - |
| *Deltaproteobacteria* | Active | 25±8 | - | - | - |
|  | Transition | 12±2 | **0.028** | - | - |
|  | Frozen | 11±2 | **0.049** | 0.976 | - |

**Fig. 1.** The richness of bacteria phyla by permafrost thawing status

---

## Author Response (AR1)

Dear Editors and Reviewers,

We thank you very much for taking time to review this manuscript. We really appreciate all your comments and suggestions! The comments and suggestions are valuable and very helpful for revising and improving our manuscript. Based on the instructions provided in your letter, we uploaded the file of the revised manuscript.

Appended to this letter is our point-by-point response to the comments raised by the reviewer. The comments are reproduced and our amended texts are in a different color.

We would like also to thank you for allowing us to submit a revised copy of the manuscript. We hope that the revised manuscript is accepted for publication in The Cryosphere

Sincerely

Weidong Kong, on behalf of all authors

**Anonymous Referee #1**

This is a technically correct manuscript on a currently relevant topic in the context of climate change and biogeochemical cycles - the response of microbes to permafrost thawing. The study shows changes in bacterial community structure and richness of drained lake basins with permafrost soil age and permafrost thawing status (active, transition and permanently frozen soil layer). In addition, there is data on soil carbon and nitrogen. The results are presented clearly and the figures are well prepared.

**Response:**

We sincerely thank reviewer #1 for the careful editing and constructive comments provided. We have carefully amended our manuscript accordingly, please see below.

Major concerns:

**Q1.** Are the samples in this study from the same soil cores as those in Kao-Kniffin et al. 2015, which is cited in the section on sampling? Kao-Kniffin et al. 2015 also describe bacterial communities with permafrost soil age and thawing status. If the soil cores are the same, please make it clear in the aims why a second analysis of bacterial communities in these samples is needed and explain what new this study adds. In any case, please take the results of Kao-Kniffin et al. 2015 into account in the discussion, especially as their conclusion (communities in active layers converge) seems to be the opposite from this manuscript (no convergence of active layer communities).

**Response:**

We appreciate the reviewer for this comment. In this manuscript, we focused on the bacterial community only, whereas Kao-Kniffin et al. 2015 investigated the community structure of the entire prokaryotes (both bacteria and archaea). For bacteria, Kao-Kniffin presented the taxonomic composition, community phylogenetic distance, and biomass. Thus, the interactive influence of permafrost age and thawing on bacterial diversity, community structure, and assembly processes still remain unexplored. Therefore, we believe further investigation is necessary and could provide essential knowledge on how permafrost age would influence the bacterial community interactively with permafrost thawing.

To address the reviewer's comments, we have added the following sentences to clarify our aims and distinguish our work from Kao–Kniffin et al.:

An earlier study at this site has revealed a high abundance of Candidatus *Methanoflorens* archaeon in the community (Kao–Kniffin et al., 2015), but how the bacteria in the permafrost of various ages would respond to thawing remains less understood. Thus, we investigated the interactive influence of permafrost thawing and age on permafrost soil bacterial community.

Following sentences are added to discuss the inconsistency between our work and Kao–Kniffin et al on community convergence:

Our results demonstrated that bacterial community structure did not converge due to permafrost thawing, as reflected by the non-significant difference in sample heterogeneity among the various permafrost layers (Supplementary Fig. 3, Supplementary Table 11). This contradicts previous studies (Deng et al., 2015; Yuan et al., 2018) in the Arctic, but was consistent with Mackelprang (2011). Our results also contradict to Kao–Kniffin et al. (2015), which reported lowered prokaryotic community differences in the active layer than in the transition and permanently frozen permafrost. Several reasons could cause this inconsistency. Firstly, different microbial communities were targeted. Kao–Kniffin et al. (2015) focused on archaeal community, whereas only bacteria were targeted in the present study. Kao–Kniffin et al. (2015) identified a single archaeon OTU accounting for over 30% of the prokaryotic community (Fig. 3 in Kao-Kniffin et al., 2015). An early study revealed that archaea exhibited a lower community variation with increasing soil depths compared with bacteria (Frank-Fahle et al., 2014). Therefore, the community convergence observed by Kao–Kniffin et al. (2015) could be due to the influence of archaea. Furthermore, the inconsistency may be related to the different community dissimilarity metrics used. Kao–Kniffin et al. (2015) used unweighted UniFrac, which only calculates the phylogenetic closeness of OTUs, and the relative abundance is not considered. This is distinctively different from the Bray-Curtis dissimilarity used in the present study, and it has been reported that unweighted and weighted community metrics examine different features of community strcture (Lozupone et al., 2011).

**Q2.** I am concerned that the connection of soil layers to thawing status is too simplified and does not take into account variation in the soil profile. Was the soil structure/chemical composition of the profiles homogeneous with depth? The description of organic layer on l. 94-96, Fig. 1 and Kao-Kniffin et al. 2015 and Mueller et al. 2015 cited in the manuscript suggest they were not. In this case, the differences in bacterial communities between soil layers cannot be directly interpreted as a thawing response (l.37., l.318), because the state of the system before thawing is not known and the differences between the layers can be due differences in other soil properties (for example organic vs. mineral layer). It is possible to compare the active, transition and frozen layers with permafrost age but that seems to have already been done by Kao-Kniffin et al. 2015? In any case, the issue of other differences between the soil layers than thawing status should be better taken into account in the manuscript. Do soil carbon and nitrogen explain the community changes?

**Response:**

We appreciate the reviewer to raise this concern. As the reviewer has pointed out, the physicochemical properties of the soils in different permafrost depths are not homogenous. Hence, the response of microbial community to thawing could be the collective effects of both thawing and the environmental differences. However, microbial transformation (as a result of permafrost thawing) would substantially change the quantity and composition of organic compounds (Mueller et al. 2015). Thus, soil physicochemical properties and bacterial community structure are interactive, and we have to admit that the individual influence would be very difficult to be disentangled. Nevertheless, Mondav et al. (2017) reported that permafrost thawing has a stronger influence on microbial community structure than soil depth. To address the reviewer's concern and emphasize the importance of environmental heterogeneity in the different permafrost layers, we have added the following paragraph to the manuscript:

Bacterial community structure in the active layer is more similar to the transition layer than to the permanently frozen layer (Fig. 3). This is consistent with those observed in other Arctic permafrost (Monteux et al, 2018, Deng et al., 2015), confirming that thawing can homogenize bacterial community structure of different soil depths. However, significant differences in bacterial community were still observed between the active and transition layers (Supplementary Table 8), instead of being identical (Monteux et al, 2018). This could be due to physiochemical heterogeneity between the soils in the different permafrost layers (Fig. 1, Kao-Kniffin, et al., 2015, Mueller et al. 2015). Thus, variations in the measured (such as TN) and unmeasured physicochemical properties (such as pH) among the different permafrost layers also contributed to the bacterial community heterogeneity and led to the significantly different bacterial communities observed.

We have added the following sentence to the conclusion section, which demands further investigation to identify the factors (both environmental and historical) that caused the distinct microbial responses to thawing.

Further studies are required to identify the environmental and historical factors that lead to the distinct responses of bacteria in the permafrost of different ages.

**Minor comments:**

**Q3.** 82-91 Please indicate where your replicate samples come from and how many there are. Here it is mentioned that there are four age classes and one soil core per age class, but the figures show a lot more data points (over 40?). Table S6: I am confused how mean relative abundance can be over 100%. Could you clarify? Please also check the definition of SD (should be standard deviation?).

**Response:**

We thank the reviewer for this comment, the following sentence has been added to clarify the number of replicates used.

In brief, 16 soil cores were collected along a chronosequence of drained lake basins, spanning in age from young (< 50 years old), medium (< 300 years old), old (< 3,000

years old), to ancient (3,000–5,000 years old) in April 2010.

And

For each permafrost age-layer combination, there were four sampling replicates.

For Table S6, we apologize for the mistake. The number presented is the bacterial richness (i.e., number of OTUs observed), but not a percentage number. The spelling of S.D. is corrected throughout the manuscript.

Original table

[revised manuscript text omitted]

**Anonymous Referee #2**

This manuscript reports significant influences of both permafrost thawing and age on bacterial richness and community structure. It also documented that permafrost thawing increased the contribution of determinism to bacterial community assembly, but didn't lead to community convergence. The study then showed that permafrost thawing had a greater influence on bacterial community than permafrost age. They extrapolate their findings to highlight that permafrost thawing in different ages can lead to distinct bacterial community compositions and different soil organic carbon degradation processes. The manuscript is well organized and figures are well prepared. I have several major concerns about this manuscript:

**Response:**

We sincerely thank reviewer #2 for the careful editing and constructive comments provided. We have carefully amended our manuscript accordingly, please see below.

**Q1**. I am not an expert in permafrost. But I noticed that the samples used in this study should be the same samples reported in Kao-Kniffinet al. 2015. In this study, the permafrost age of different samples had been measured by Kao-Kniffin et al. 2015. However, basin age is used in that study instead of permafrost age. Does it mean that basin age is equal to permafrost age? If so, Why did this study reported different soil total organic carbon and total nitrogen from Kao-Kniffin et al. 2015?

**Response:**

The "basin age" used by Kao-Kniffin et al. 2015 was obtained from Hinkel et al., 2003. This age is calculated using radiocarbon dating on sample at the interface of the lacustrine sediment and the *in situ* peat, which represents the point in time of lake drainage and revegetation of the basin surface. The basin drainage is associated with vegetation establishment, organic matter accumulation, and ice-wedge growth below the drained lake basin (Hinkel et al., 2003). Therefore, we used the basin age as a proxy for the formation of permafrost (permafrost age), and explored the influence of permafrost age on the response of bacteria to permafrost thawing. We did not use the term basin age, as we feel that permafrost age is more meaningful than basin age in this case. We apologize for not stating this clearly, and following sentences are now added to clarify this.

Amended manuscript:

Approximately 20% of the Arctic coastal plains of northern Alaska contain thaw lakes drained at various stages since the mid-Holocene, which were then developed into ice-rich permafrost (Hinkel et al., 2003). These drained lake basins contain soils ranging from freshly developed organic layers on sediments to fully developed ancient permafrost soils (Mueller et al., 2015). By using the drained thaw lake basin age as a proxy for the time of permafrost formation, it provides an opportunity to investigate the influence of permafrost age on microbial community during permafrost degradation. An earlier study at this site has revealed a high abundance of Candidatus *Methanoflorens* archaeon in the community (Kao–Kniffin et al., 2015), but how the bacteria in the permafrost of various ages would respond to thawing remains less understood. Thus, we investigated the interactive influence of permafrost thawing and age on permafrost soil bacterial community.

Regarding the inconsistent environment factors with Kao-Kniffinet al. 2015, the environmental factors (SOC and TN) were independently measured in the present study. They are not substantially different from those in Kao-Kniffinet al. 2015 (Table 1), and the differences could be related to sample storage and the equipment used.

Table 1 Comparison of soil organic carbon (SOC) and total nitrogen (TN) using in the present study and Kao-Kniffin et al.

| | SOC (%) | | TN (%) | |
| --- | --- | --- | --- | --- |
| | The present study | Kao-Kniffin et al. 2015. | The present study | Kao-Kniffin et al. 2015. |
| Basin age | | | | |
| Young | 8.8 (2.0) | 6.9 (1.9) | 0.43 (0.09) | 0.43 (0.11) |
| Medium | 9.8 (2.5) | 8.4 (1.6) | 0.45 (0.11) | 0.49 (0.09) |
| Old | 12.4 (2.2) | 13.4 (1.6) | 0.57 (0.10) | 0.77 (0.09) |
| Ancient | 13.3 (3.2) | 10.7 (2.5) | 0.60 (0.13) | 0.62 (0.14) |
| Soil depth layer | | | | |
| Active | 16.9 (2.1) | 14.1 (2.1) | 0.73 (0.09) | 0.78 (0.11) |
| Transition | 10.4 (2.1) | 9.6 (1.8) | 0.50 (0.09) | 0.57 (0.11) |
| Permafrost | 5.6 (1.0) | 5.8 (1.4) | 0.29 (0.05) | 0.36 (0.09) |

The values indicate means and standard error (in parentheses)

**Q2.** It seems that Kao-Kniffin et al. 2015 also used amplicon sequencing of 16S rRNA gene to analyze bacterial communities in different permafrost ages and thawing status. They found that community composition appeared to converge in the active layer, however, the authors in this study didn't observe the community convergence due to permafrost thawing. Can you explain why you reanalyzed bacterial communities of these samples? At least, please compare your study with the results of Kao-Kniffin et al. 2015 and provide more discussion.

**Response:**

We appreciate the reviewer for this comment. In this manuscript, we focused on the bacterial community only, whereas Kao-Kniffin et al. 2015 investigated the community structure of the entire prokaryotes (both bacteria and archaea). For bacteria, Kao-Kniffin presented the taxonomic composition, community phylogenetic distance, and biomass. Thus, the interactive influence of permafrost age and thawing on bacterial diversity, community structure, and assembly processes still remain unexplored. Therefore, we believe further investigation is necessary and could provide essential knowledge on how permafrost age interact with permafrost thawing. To address the reviewer's comments, we have added the following sentences to clarify our aims and distinguish our work from Kao–Kniffin et al.:

An earlier study at this site has revealed a high abundance of Candidatus *Methanoflorens* archaeon in the community (Kao–Kniffin et al., 2015), but how the bacteria in the permafrost of various ages would respond to thawing remains less understood. Thus, we investigated the interactive influence of permafrost thawing and age on permafrost soil bacterial community.

Following sentences are added to discuss the inconsistency between our work and Kao–Kniffin et al on community convergence:

Our results demonstrated that bacterial community structure did not converge due to permafrost thawing, as reflected by the non-significant difference in sample heterogeneity among the various permafrost layers (Supplementary Fig. 3, Supplementary Table 11). This contradicts previous studies (Deng et al., 2015; Yuan et al., 2018) in the Arctic, but was consistent with Mackelprang (2011). Our results also contradict to Kao–Kniffin et al. (2015), which reported lowered prokaryotic community differences in the active layer than in the transition and permanently frozen permafrost. Several reasons could cause this inconsistency. Firstly, different microbial communities were targeted. Kao–Kniffin et al. (2015) focused on archaeal community, whereas only bacteria were targeted in the present study. Kao–Kniffin et al. (2015) identified a single archaeon OTU accounting for over 30% of the prokaryotic community (Fig. 3 in Kao-Kniffin et al., 2015). An early study revealed that archaea exhibited a lower community variation with increasing soil depths compared with bacteria (Frank-Fahle et al., 2014). Therefore, the community convergence observed by Kao–Kniffin et al. (2015) could be due to the influence of archaea. Furthermore, the inconsistency may be related to the different community dissimilarity metrics used. Kao–Kniffin et al. (2015) used unweighted UniFrac, which only calculates the phylogenetic closeness of OTUs, and the relative abundance is not considered. This is distinctively different from the Bray-Curtis dissimilarity used in the present study, and it has been reported that unweighted and weighted community metrics examine different features of community strcture (Lozupone et al., 2011).

**Q3.** The thickness of active, transition and permafrost layers should be different in young, medium, old and ancient permafrost. Please provide more information about the soil profile of different layers in four kinds of permafrost. More variables should be taken into account to undermining the mechanism of bacterial response to permafrost thawing in different permafrost age. I'm not sure that structural equation modelling is a good method to quantify the relative importance of permafrost thawing status and age on bacterial community without any other environmental variables. Please incorporate more variables in structural equation modelling to show how permafrost thawing status and age influenced bacterial community directly or indirectly

**Response:**

We appreciate the reviewer for this comment. We agree with the reviewer that soil profile characteristics are important in determining the response of bacteria to permafrost thawing. Unfortunately, we have only obtained a small quantity of soil samples, and this does not allow us to measure soil properties in great detail. Therefore, we decided to focus on TOC and TN, which are known to be substantially different in the permafrost with different ages and thawing statuses (Mueller et al., 2015). We have also amended the manuscript to acknowledge the limitation of our work and propose further works that are required to identify the environmental factors that shape bacterial response to permafrost thawing.

Amended manuscript (method section)

Due to sample quantity limitation, two of the most important soil physicochemical properties: total organic carbon (TOC) and total nitrogen (TN) were measured using dry combustion (Vario MAX CNS Analyzer, Elementar, Hanau, Germany). These factors have been reported to substantially vary in samples with different permafrost ages and thawing statuses (Mueller et al., 2015). For other soil properties and soil profile descriptions please see Kao-Kniffin et al (2005).

To address the reviewer's concern and emphasize the importance of environmental heterogeneity in different permafrost layers, we have added the following paragraph to the manuscript:

Amended manuscript (discussion section)

Bacterial community structure in the active layer is more similar to the transition layer than to the permanently frozen layer (Fig. 3). This is consistent with those observed in other Arctic permafrost (Monteux et al, 2018, Deng et al., 2015), confirming that thawing can homogenize bacterial community structure of different soil depths. However, significant differences in bacterial community were still observed between the active and transition layers (Supplementary Table 8), instead of being identical (Monteux et al, 2018). This could be due to physiochemical heterogeneity between the soils in the different permafrost layers (Fig. 1, Kao-Kniffin, et al., 2015, Mueller et al. 2015). Thus, variations in the measured (such as TN) and unmeasured physicochemical properties (such as pH) among the different permafrost layers also contributed to the bacterial community heterogeneity and led to the significantly different bacterial communities observed.

Amended manuscript (Conclusion section)

Further studies are required to identify the environmental and historical factors that lead to the distinct responses of bacteria in the permafrost of different ages.

Using ranked soil age in SEM to investigate the influence of pedogenic development has been used by Laliberté et al., 2017. Here, we use this idea to investigate the influence of permafrost age on the response of bacteria to permafrost thawing. We repeated the SEM analysis incorporating the TOC and TN as suggested by the reviewer, and a large proportion of the richness and community structure were still explained by the permafrost age and thawing status. Therefore, permafrost age and thawing status can influence the richness and community structure via other unmeasured environmental factors or historical effects. The present study does not attempt to identify the environmental factors that are different in the soils of different permafrost ages and thawing statuses. Instead, we tried to raise the awareness that different aged permafrost may response to climate change induced permafrost thawing differently. To address this comment, we incorporate TOC and TN into the SEM analysis, and performed Random Forest analysis to identify the importance of permafrost age and thawing status. We also amended the results and conclusion sections to address the needs of identifying environmental/historical factors that directly cause the different responses of bacteria in the permafrost of various ages.

Amended manuscript (Results section)

Structural equation modeling (SEM) revealed that both permafrost thawing status and age significantly contributed to bacterial richness. Permafrost thawing status exhibited a higher contribution than age (standard regression weight of 0.51 and -0.30, respectively, both $P < 0.05$) to bacterial richness (Fig. 4a). However, the influences of TOC and TN on bacterial richness were not detected. This is consistent with the Random Forest analysis results, which only identified permafrost thawing and age as the significant determinants of bacterial richness (Supplementary Fig. 5). For community structure, permafrost thawing exhibited an indirect influence on NMDS1 via TN (standard regression weight of 0.58 and -0.63, both $P < 0.001$, Fig. 4b). In comparison, both permafrost age and thawing status significantly contributed to NMDS2 (standard regression weight of -0.34 and 0.59, respectively, both $P < 0.01$), while TN also exhibited a significant influence on NMDS2 (-0.49, $P = 0.002$). The significant contributions of TN, permafrost thawing, and age were consistently identified using the Random Forest approach (Supplementary Fig. 5).

Amended manuscript (Conclusion section)

Further studies are required to identify the environmental and historical factors that lead to the distinct responses of bacteria in the permafrost of different ages.

Minor comments:

**Q4.** Fig.2 Can you provide information about bacterial phylogenetic diversity of bacteria in different permafrost age and thawing status?

**Response:**

We appreciate the reviewer for this comment. We have now included the results of

Phylogenetic diversity and Shannon diversity in the results and discussion sections.

Amended manuscript (Results section)

In comparison, the influence of permafrost age and thawing status on bacteria Shannon diversity was non-significant (Two-way ANOVA, $P = 0.058$ and $0.53$, respectively, Supplementary Fig. 2). This contrastively differed from the phylogenetic diversity, where significant influence was observed for age ($P = 0.015$) and thawing ($P = 0.001$).

Amended manuscript (Discussion section)

Furthermore, the phylogenetic diversity exhibited a greater sensitivity to permafrost thawing than the Shannon diversity (Supplementary Fig. 2). As phylogenetically close-related microorganisms have similar habitat associations, phylogeny-based community metrics could infer potential community functional change (Stegen et al., 2012). Hence, this suggests that community function could be more sensitive to permafrost thawing than community composition.

**Q5.** Fig.4 the path coefficients in structural equation modelling can indicate the positive or negative correlations between two variables. Therefore, the raw value should be shown here, instead of the absolute value.

**Response:**

We appreciate the reviewer for this comment. We have now included the signs of path coefficients in both the figure and manuscript text.

Amended manuscript

[Figure]

Amended manuscript (results section)

[revised manuscript text omitted]

 Fig. 1

[Figure]

[Figure]

Fig. 3

[Figure]

Fig. 4

[Figure]

A

Permafrost thawing    Permafrost age

-0.3*          0.51***

Richness

$\chi^2$=0.183, df=1, *P*=0.669,
GFI=0.997
CFI=1.0
NFI=0.987
RMSEA=0.000, *P*=0.686

B

Permafrost Thawing    Permafrost age

0.58***        0.59***    -0.34**

TN

-0.63***    --0.49**

NMDS1        NMDS2

$\chi^2$=5.902, df=5, *P*=0.316
GFI=0.946
CFI=0.958
NFI=0.813
RMSEA=0.064, *P*=0.379

*      *P*<0.05
* *   *P*<0.01
*** *P*<0.001